

# Childhood maltreatment and suicidal ideation in Chinese children and adolescents: the mediation of resilience

Xue Chen[1,2,*], Linling Jiang[3,*], Yi Liu[3], Hailiang Ran[1], Runxu Yang[3], Xiufeng Xu[3], Jin Lu[3] and Yuanyuan Xiao[1]

[1] School of Public Health, Kunming Medical University, Kunming, China
[2] The Second Affiliated Hospital, Kunming Medical University, Kunming, China
[3] The First Affiliated Hospital, Kunming Medical University, Kunming, China
[*] These authors contributed equally to this work.

Corresponding authors
Jin Lu, Jinlu2000@163.com
Yuanyuan Xiao, 33225647@qq.com

## ABSTRACT

**Background**. Childhood maltreatment could increase the risk of suicidal ideation (SI) in adolescents. However, the mediation of resilience in this association remains unclear.
**Methods**. A population-based cross-sectional study has been done among 3,146 Chinese adolescents. We collected relevant information from the study participants by using self-administered questionnaire. Chinese version of the Childhood Trauma Questionnaire (CTQ), the Resilience Scale for Chinese Adolescents (RSCA), and the Beck Scale for Suicide Ideation (BSSI) were used to measure childhood maltreatment, resilience, and SI, respectively. Univariate and multivariate binary Logistic regression models were employed to estimate crude and adjusted associations between childhood maltreatment, resilience, and SI. Path analysis has subsequently been performed to measure the mediation of resilience in this association.
**Results**. Multivariate Logistic regression models revealed that compared to non-abused counterparts, adolescents who had ever experienced any type of childhood maltreatment was 1.74 times likely to report SI. Among the specific types of childhood maltreatment, emotional abuse showed the strongest association with SI (adjusted OR = 3.01, 95% CI [2.37–3.82]). Path model suggested that over one-third (39.8%) of the total association between childhood maltreatment and SI was mediated via resilience. Emotion regulation and interpersonal assistance were the most prominent mediators among all dimensions of resilience.
**Conclusions**. Resilience played as a significant mediator in the association between childhood maltreatment and SI. Resilience-oriented intervention measures could be considered for suicidal risk prevention among abused Chinese adolescents.

## INTRODUCTION

Suicide is one leading cause of death among adolescents worldwide (*WHO, 2018*). The suicidality model emphasizes the continuous developmental stages in sequence: suicidal ideation (SI), suicide plan, suicide attempt, and finally, completed suicide (*Sveticic & De Leo, 2012*). Based on this theory, suicidal risk evolves from low to high with the

progression along this path (*Szanto et al., 2003*). From this point of view, SI can be seen as an imminent precursor of the subsequent suicide action, thus identifying suicide ideators is critical for effective suicide intervention.

SI can be influenced by multiple indicators among adolescents, like age, gender, mental disorders, educational levels, social support, interpersonal difficulties, and family conflict (*Ahorsu et al., 2020*; *Salama et al., 2020*; *Xiao et al., 2019*). In addition, a large body of studies suggested that childhood maltreatment is a salient risk factor for SI (*Falgares et al., 2018*; *Stickley et al., 2020*). Childhood maltreatment is defined as the abuse or neglect of children under the age of 16 (*Centers for Disease Control and Prevention (CDC), 2017*). It has been corroborated that childhood maltreatment can significantly increase future risk of affective disorders, such as depression, anxiety, borderline personality disorder, behavioral problems, or even suicide (*Rehan et al., 2017*; *Rafiq, Campodonico & Varese, 2018*; *Fry, McCoy & Swales, 2012*).

Childhood maltreatment is also a concern in China. Previous studies have revealed that 36.6% of the Chinese population reported physical child abuse experience (*Ji & Finkelhor, 2015*), the prevalence of childhood sexual abuse was 24.8% and 17.6% for female and male college students (*Chen et al., 2010*). Although the positive association between childhood abuse and SI has been well established, mechanism underneath this association, specifically, possible mediators in this association, remains unclear.

In the field of psychology, resilience has been defined as the ability to thrive in hardships or to recover from negative events (*De Terte & Stephens, 2014)*). Adolescent resilience is a multi-layer concept, which contains characteristics from individual level (such as goal concentration, emotion regulation, positive perception), family level (such as family support), and social environment level (such as interpersonal assistance) (*Olsson et al., 2003*). Prior studies suggested that children with childhood maltreatment experience had significantly lower psychological resilience (*Dubowitz et al., 2016*); meanwhile, it has been corroborated that psychological resilience level was significantly associated with the occurrence of SI (*Cong et al., 2019*). Therefore, it is logical to suspect that resilience may play as a mediator in the association between childhood maltreatment and SI. However, to our best knowledge, existing literature did not thoroughly investigate this hypothesis among Chinese adolescents.

Aiming to address this issue in this cross-sectional study, we intended to estimate the possible mediation of resilience in the association between childhood maltreatment and SI in a large group of Chinese adolescents. We put forward two hypotheses: (1) Different types of childhood maltreatment experiences are significantly but discordantly associated with increased risk of SI; (2) Resilience exerts prominent mediation in the association between childhood maltreatment and SI.

## MATERIALS & METHODS

### Participants

A cross-sectional survey was conducted in Lincang, western China Yunnan province from December 1 to December 13, 2019. Participants were selected using a multistage simple

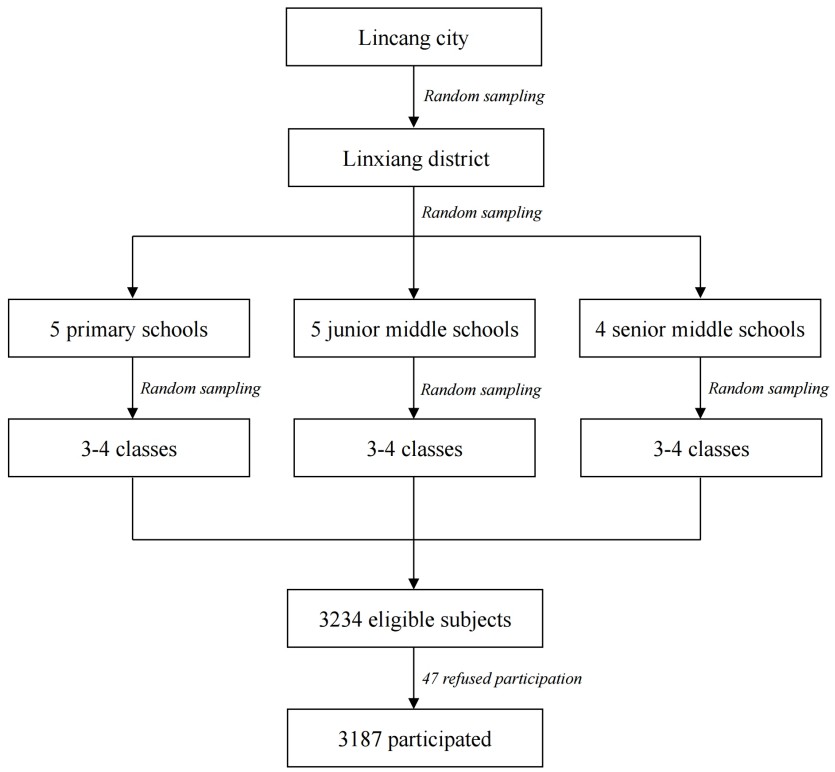

**Figure 1** **The process for sampling and study participants selection.**

random cluster sampling method. In stage one, we randomly chose Linxiang district from a total of eight districts within Lingcang; in stage two, five primary schools, five junior middle schools, and four senior middle schools were selected randomly; at last, three to four classes were randomly selected in each chosen school based on the required sample size. All eligible students in these classes were preliminarily included. The process for sampling and study participants selection was shown in Fig. 1. The questionnaires were self-administered by respondents. When filling in the questionnaires, any confusion of the respondents can be consulted immediately at sites. To avoid any potential information lost, each completed questionnaire was also carefully reviewed immediately by pre-trained quality control personnel.

Adolescents aged above 10 and below 18 were eligible study subjects, the rationale for a lower limit of 10 was based on the finding that a child cannot well understand the concept of suicide until the age of 10 (*Mishara, 1999*). Children and adolescents were further screened by using the following exclusion criteria: (1) Illiterate; (2) Physically ill, cannot finish the survey; (3) Auditory dysfunction or language disorder; (4) Unconscious or delirious, cannot clearly express oneself. The presence of criteria (2)–(4) had been carefully evaluated by professional clinicians deployed at the survey sites. Prior to the survey, written consents were obtained from both the participants and their legal guardians. The study protocol
was reviewed and approved by the Third People's Hospital of Lincang Ethics Committee (Approval number: 2019-01).

## Measurements
### General characteristics

General characteristics of the participants, mainly demographics (gender, age, ethnicity, residence, grade, study style) and socioeconomic status (father's age, mother's age, father's education level, mother's education level, family income, marital status of the parents), were collected by using a self-developed questionnaire.

### Childhood maltreatment

Chinese version of the Childhood Trauma Questionnaire (CTQ), a self-report scale of 28 items, was used to assess childhood physical abuse (PA), emotional abuse (EA), sexual abuse (SA), physical neglect (PN), and emotional neglect (EN). A 5-point scale rated answers are used for each question: 1 = never, 2 = rarely, 3 = sometimes, 4 = often, and 5 = very often. The total score for each type of maltreatment ranges between 5 and 25. The following recommended cut-offs were used to dichotomize study subjects: 8 for PA, 9 for EA, 6 for SA, 8 for PN, 10 for EN (*Bernstein et al., 1994*). The Chinese version of CTQ showed acceptable reliability (*Zhao et al., 2005*).

### SI

We used Chinese version of the Beck Scale for Suicide Ideation (BSSI) to assess lifetime SI of the participants (*Beck, Steer & Ranieri, 1988*). BSSI is one of the few suicide assessment instruments that possess predictive validity for completed suicide (*Zhang & Brown, 2007*). Participants with one of the following two conditions were defined as suicide ideator: (1) Answered "Weak" or "Moderate to strong" to the question "Desire to make active suicide attempt"; (2) Answered "Would leave life/death to chance" or "Would avoid steps necessary to save or maintain life" to the question "Passive suicidal desire" (*Xiao et al., 2019*).

### Resilience

The 27-item Resilience Scale for Chinese Adolescents (RSCA) was used to evaluate the five dimensions of resilience (goal concentration, emotion regulation, positive perception, family support, and interpersonal assistance). RSCA was designed by *Hu & Gan (2008)*, and its reliability and validity have been corroborated in Chinese adolescents. The higher combined RSCA score reflects a higher level of resilience. The Cronbach's $\alpha$ for RSCA based on our analytical sample was 0.82 (Bootstrap 95% CI [0.80–0.83]).

## Statistical analysis

Descriptive statistics were used to describe and compare general characteristics. We used univariate and multivariate binary Logistic regression models to explore the crude and adjusted associations between childhood maltreatment, resilience, and SI. At first, univariate model was used to screen for prominent demographic and socioeconomic covariates of SI at a lower significance level of 0.10. Prominent variables discerned by univariate models were further included into the subsequent multivariate models.

Childhood maltreatment and resilience were analyzed separately in model 1 and 2, then collectively in model 3. Based on the results of univariate and multivariate models, hypothetical path models were constructed to evaluate the mediation of resilience in the association between childhood maltreatment and SI.

All analyses were performed by using the R software (Version 3.6.2). Possible inter-correlation which caused by clustering sampling design was adjusted for by using R analytical packages for survey data, such as "survey" and "lavaan. survey". The significance level was set as $p < 0.05$, two-tailed.

## RESULTS

### General features of study participants

A total of 3,234 eligible adolescents were identified by using the inclusion and exclusion criteria, 47 students cannot participate because they asked for sick leave, with a survey response rate of 98.5%. Among the respondents, 3,146 provided valid and complete information, and the effective response rate was 97.3%. Characteristics of all included participants were displayed in Table 1. A total of 1,091 adolescents reported SI, accounted for 34.7% (95% CI [28.4%–48.0%]). Besides, a higher prevalence of SI was observed in girls. CTQ scores for all five types of childhood maltreatment were significantly higher for suicidal ideators ($p < 0.01$). The mean of combined resilience score was 93.9 for all participants, and suicidal ideators showed lower scores in all five dimensions of resilience ($p < 0.01$).

### Associations between childhood maltreatment, resilience, and SI

Based on the cut-offs provided above for different dimensions of CTQ, 2,181 adolescents reported at least one type of childhood maltreatment, accounting for 69.3% (95% CI [61.9%–76.0%]). We used the median of RSCA (94) to dichotomize study participants as no commonly used cut-off for RSCA has been recommended. Gender, age, grade, study style, ages of both parents, education level of father, and marital status of the parents were significant covariates included into the subsequent multivariate models. The fitting results of the final multivariate model (Multivariate 3 in Table 2) indicated that: grade, study style, marital status of the parents were significantly associated with SI, participants of higher grade, were day students, whose parents were divorced, re-married, or widowed were observed increased risk of SI. Moreover, after adjusted for prominent covariates, for adolescents who had ever experienced childhood maltreatment, the odds of SI was 1.91 times (95% CI [1.51–2.41]) compared to non-abused adolescents; participants with a higher level of resilience (RSCA score $\geq$ 94) were 80% less likely (95% CI [74%–84%]) to report SI than participants with a lower level of resilience (RSCA score < 94).

Considering of the positive relationship between childhood maltreatment and SI, we further investigated the associations between different types of child maltreatment and SI, and the results were summarized in Fig. 2: all types of maltreatment were associated with prominently increased odds of SI, particularly, adolescents who had experienced emotional abuse showed the highest SI risk (adjusted OR = 3.01, 95% CI [2.37–3.82]), followed by emotional neglect (adjusted OR = 1.78, 95% CI [1.35–2.36]), sexual abuse (adjusted OR
**Table 1  Major characteristics of the study participants, Lincang, China, 2019.**

| Features | SI | Non-SI | All |
|---|---|---|---|
| Gender (*N*, %): Girls | 697 (63.89) | 1,012 (49.25) | 1,709 (54.32) |
| Age (Mean (SE)) | 13.74 (0.06) | 13.09 (0.05) | 13.31 (0.04) |
| Ethnicity (*N*, %) | | | |
|     Han | 728 (66.73) | 1,384 (67.35) | 2,112 (67.13) |
|     Yi | 127 (11.64) | 238 (11.58) | 365 (11.60) |
|     Dai | 82 (7.52) | 128 (6.23) | 210 (6.68) |
|     Other | 154 (14.12) | 305 (14.84) | 459 (14.59) |
| Residence (*N*, %) | | | |
|     Township | 491 (45.00) | 1,026 (49.93) | 1,517 (48.22) |
|     Village | 600 (55.00) | 1,029 (50.07) | 1,629 (51.78) |
| Grade (*N*, %) | | | |
|     Primary school | 300 (27.50) | 832 (40.49) | 1,132 (35.98) |
|     Junior high school | 378 (34.64) | 691 (33.63) | 1,069 (34.00) |
|     Senior high school | 413 (37.86) | 532 (25.89) | 945 (30.04) |
| Study style (*N*, %) | | | |
|     Day students | 368 (33.73) | 913 (44.43) | 1,281 (40.72) |
|     Boarding students | 723 (66.27) | 1,142 (55.57) | 1,865 (59.28) |
| Father's age (Mean (SE)) | 42.76 (0.18) | 42.02 (0.13) | 42.27 (0.10) |
| Mother's age (Mean (SE)) | 39.94 (0.16) | 39.25 (0.11) | 39.49 (0.09) |
| Father's education level (*N*, %) | | | |
|     Elementary school and below | 343 (31.44) | 542 (26.37) | 885 (28.13) |
|     Junior high school and above | 633 (58.02) | 1,299 (63.21) | 1,932 (61.41) |
|     Missing or unknown | 115 (10.54) | 214 (10.41) | 329 (10.46) |
| Mother's education level (*N*, %) | | | |
|     Elementary school and below | 401 (36.76) | 676 (32.90) | 1,077 (34.23) |
|     Junior high school and above | 617 (56.55) | 1,199 (58.35) | 1,816 (57.72) |
|     Missing or unknown | 73 (6.69) | 180 (8.76) | 253 (8.04) |
| Marital status of the parents (*N*, %) | | | |
|     Married | 909 (83.32) | 1,797 (87.45) | 2,706 (86.01) |
|     Divorced | 87 (7.97) | 130 (6.33) | 217 (6.90) |
|     Re-married | 62 (5.68) | 78 (3.80) | 140 (4.50) |
|     Widowed | 33 (3.02) | 47 (2.29) | 80 (2.54) |
| Family income (*N*, %) | | | |
|     Stable | 981 (89.92) | 1,926 (93.72) | 2,907 (92.40) |
|     Unstable | 108 (9.90) | 128 (6.23) | 236 (7.50) |
| Childhood maltreatment (CTQ sores, Mean (SE)) | | | |
|     Emotional abuse | 8.66 (0.11) | 6.33 (0.04) | 7.13 (0.05) |
|     Physical abuse | 6.59 (0.08) | 5.82 (0.04) | 6.09 (0.04) |
|     Sexual abuse | 5.54 (0.05) | 5.22 (0.02) | 5.33 (0.02) |
|     Emotional neglect | 11.75 (0.15) | 9.43 (0.09) | 10.23 (0.08) |
|     Physical neglect | 8.03 (0.09) | 6.83 (0.05) | 7.25 (0.05) |

**Table 1** (*continued*)

| Features | SI | Non-SI | All |
|---|---|---|---|
| Resilience (RSCA scores, Mean (SE)) | | | |
|     Combined score | 84.29 (0.46) | 98.96 (0.34) | 93.87 (0.30) |
|     Goal concentration (Dimension 1) | 15.48 (0.14) | 18.12 (0.11) | 17.21 (0.09) |
|     Emotional regulation (Dimension 2) | 17.58 (0.16) | 23.02 (0.11) | 21.14 (0.10) |
|     Positive perception (Dimension 3) | 13.20 (0.11) | 13.78 (0.09) | 13.58 (0.07) |
|     Family support (Dimension 4) | 19.67 (0.11) | 21.36 (0.08) | 20.78 (0.06) |
|     Interpersonal assistance (Dimension 5) | 18.36 (0.17) | 22.67 (0.11) | 21.18 (0.10) |

Notes.
CTQ, Childhood Trauma Questionnaire; SI, Suicidal ideation; RSCA, the combined score of Resilience Scale for Chinese Adolescents.

**Table 2  Results of multivariate regression models for associated factors of SI.**

| Covariates | Multivariate 1 OR (95% CI) | Multivariate 2 OR (95% CI) | Multivariate 3 OR (95% CI) |
|---|---|---|---|
| Age (+1 year) | 1.12 (1.04, 1.20) | 1.12 (1.04, 1.21) | 1.11 (1.02, 1.21) |
| Father's age (+5 years) | 1.00 (0.85, 1.19) | 1.04 (0.87, 1.25) | 1.02 (0.85, 1.23) |
| Mother's age (+5 years) | 1.08 (0.93, 1.25) | 1.06 (0.93, 1.21) | 1.09 (0.95, 1.26) |
| Gender (Ref: Boys) | | | |
|     Girls | 2.09 (1.76, 2.47) | 1.96 (1.67, 2.31) | 2.07 (1.78, 2.42) |
| Grade (Ref: Primary school) | | | |
|     Junior high school and above | 2.15 (1.37, 3.36) | 1.70 (1.12, 2.58) | 1.91 (1.16, 3.15) |
| Study style (Ref: Day students) | | | |
|     Boarding students | 0.56 (0.47, 0.67) | 0.65 (0.53, 0.80) | 0.63 (0.51, 0.78) |
| Father's education level (Ref: Elementary and below) | | | |
|     Junior high school and above | 0.96 (0.79, 1.16) | 1.07 (0.90, 1.27) | 1.10 (0.93, 1.29) |
| Marital status of the parents (Ref: Married) | | | |
|     Other | 1.48 (1.09, 2.02) | 1.62 (1.08, 2.43) | 1.52 (1.02, 2.27) |
| Childhood maltreatment (Ref: No) | | | |
|     Yes | 3.00 (2.45, 3.69) | | 1.91 (1.51, 2.41) |
| Resilience (Ref: RSCA scores <94) | | | |
|     RSCA scores ≥ 94 | | 0.17 (0.14, 0.22) | 0.20 (0.16, 0.26) |

Notes.
RSCA, the combined score of Resilience Scale for Chinese Adolescents.

= 1.68, 95% CI [1.21–2.33]), physical abuse (adjusted OR = 1.53, 95% CI [1.24–1.88]), and physical neglect (adjusted OR =1.34, 95% CI [1.14–1.56]).

## Mediation of resilience in maltreatment-SI association

Based on the previous analytical results, a hypothetical path model was put forward to illustrate the associations between childhood maltreatment, SI, and resilience. The path model justified that 39.8% of the association was mediated via resilience. Then we fitted a series of path model for 5 types of childhood maltreatment separately: after controlled for other related factors, EA, EN, and PN were indirectly associated with SI through resilience,

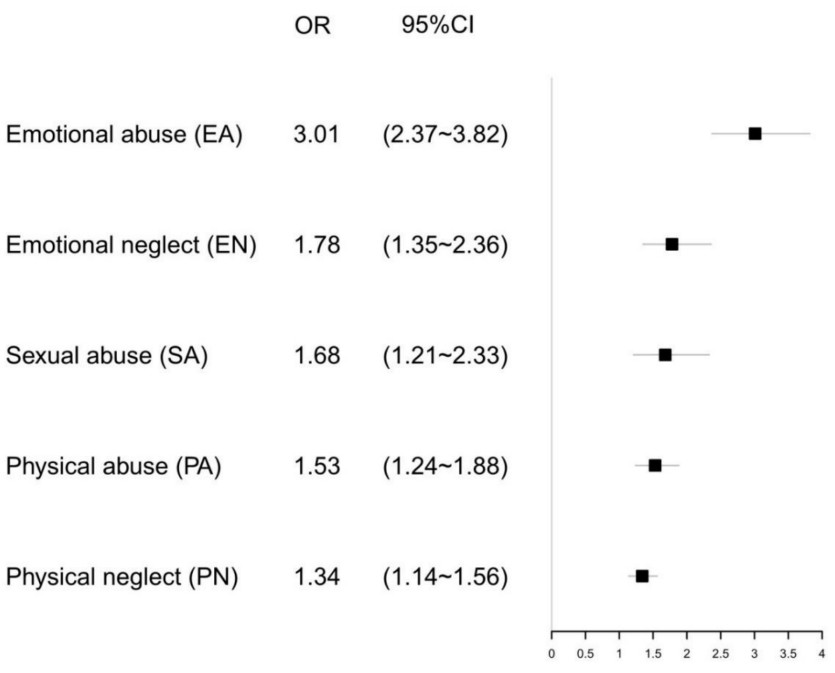

**Figure 2 Adjusted ORs with 95% CIs of SI (suicidal ideation) for types of childhood maltreatment.** Adjusted for age, gender, study style, marital status of the parents, father's education level.

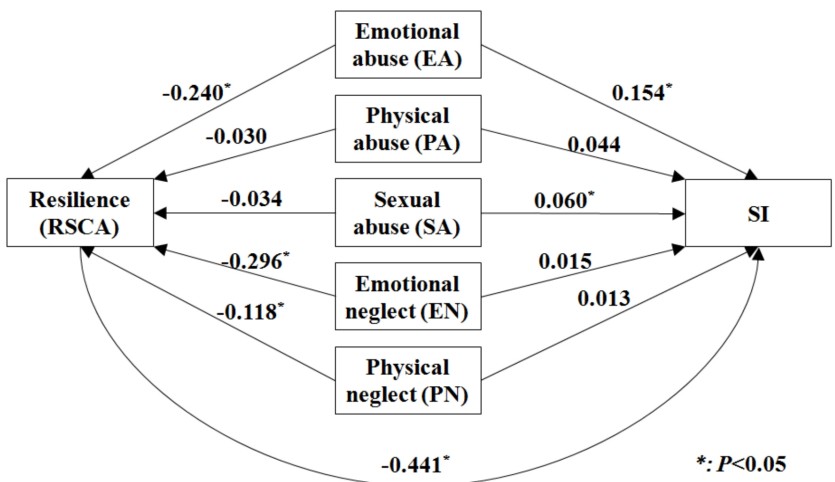

**Figure 3 The hypothetical path model of five types of childhood maltreatment, resilience and SI.**

especially for EN and PN, as their direct associations with SI became insignificant after adjusted for the mediation of resilience (Fig. 3).

We further discussed the mediation of resilience in the association between EA, EN, PN, and SI by using its dimensions. Results showed that except for positive perception, the other four dimensions of resilience were salient mediators. Emotion regulation showed the

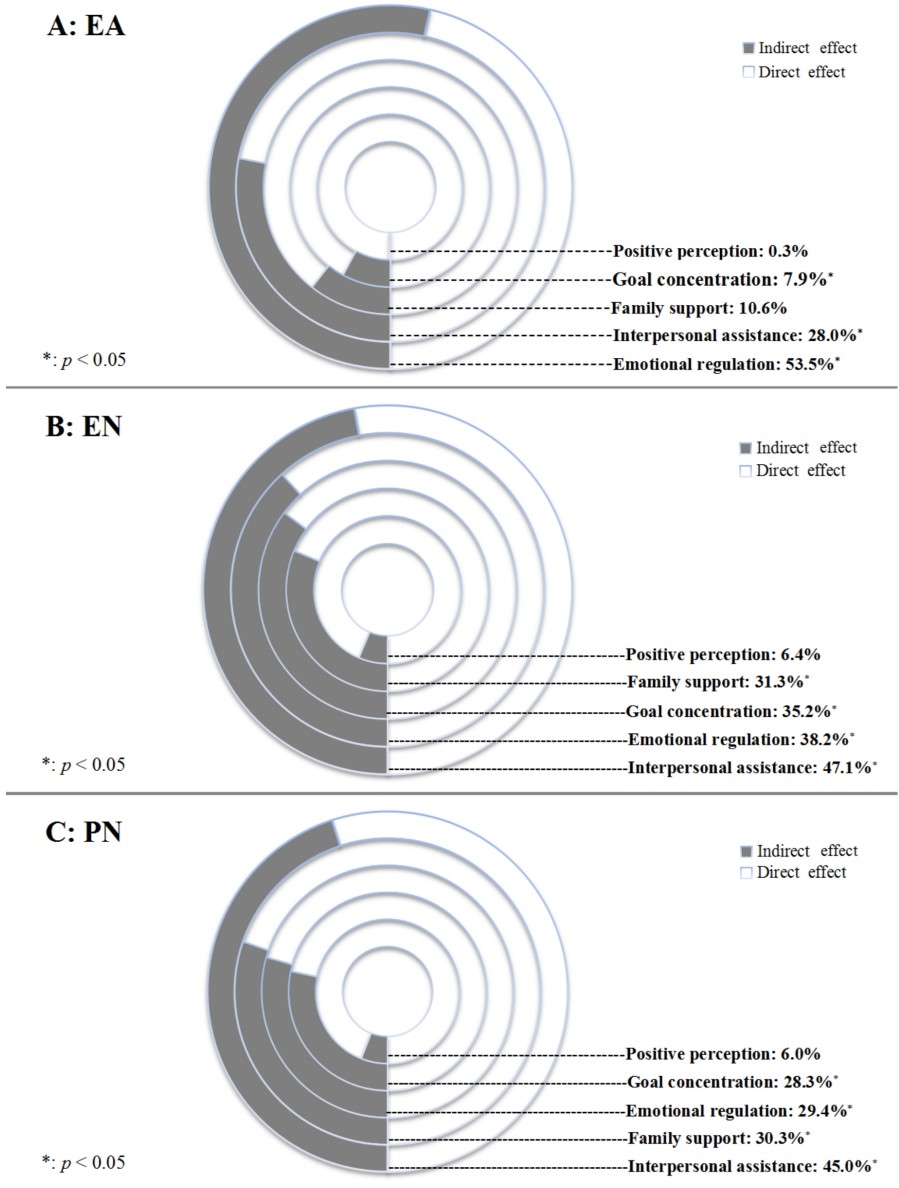

**Figure 4** **Proportion of mediating effect in association of EA, PA, PN and SI by different dimensions of resilience.** EA, Emotional abuse; PA, Physical abuse. PN: Physical neglect.

strongest mediation for individuals who had exposed to EA, whereas for EN and PN, the strongest mediator was interpersonal assistance (Fig. 4).

## DISCUSSION

In this cross-sectional study of 3,146 Chinese children and adolescents, we found a strong association between childhood maltreatment and SI, and among all types of childhood maltreatment, emotional abuse showed the strongest association with SI. More importantly, as hypothesized, resilience prominently mediated the association between childhood

maltreatment and SI: accounted for 39.8% of the total association. When dissecting resilience by using its dimensions, emotion regulation and interpersonal assistance were the most prominent mediators. The major findings of our study can help understand the intimate relationship between childhood maltreatment and SI, shed new light in the prevention of suicidal behaviors which stem from the past child abuse experiences among Chinese children and adolescents.

In the current study, we found that SI was prevalent among Chinese children and adolescents, as a lifetime prevalence of 34.7% has been found. This result was comparable to a previous study (32%) (*Tan, Xia & Reece, 2018*). Based on the analytical results, all types of childhood maltreatment were significantly and positively associated with SI. The positive connection between childhood maltreatment and SI is well supported by existing literature. First, childhood maltreatment could disturb the hypothalamic–pituitary–adrenal axis. It has been found that adult women with childhood abuse experience had higher plasma cortisol levels than non-abused individuals, and elevated cortisol is an identified risk factor for future SI (*Shalev et al., 2019*). Second, childhood maltreatment impairs cognitive function (*Miller & Esposito-Smythers, 2013*), and it has been repeatedly verified that cognitive impairment is directly associated with SI (*Pu, Setoyama & Noda, 2017*).

The results of path model revealed that over one-third of the association between childhood maltreatment and SI was inversely mediated via resilience. Although no studies on exactly the same topic can be compared with, it has been disclosed that resilience significantly mitigated suicidal risk associated with childhood trauma (*Poole, Dobson & Pusch, 2017*; *Meng et al., 2018*). Two common perspectives currently exist regarding to the concept of resilience: the trait-oriented perspective and the process-oriented perspective, and the latter is more widely accepted by researchers, for it emphasizes that resilience is a process of, or the outcome of, or the capacity for successful adaptation after setbacks. In other words, resilience can be modified. *Fergus & Zimmerman (2005)* have proposed that internal "assets" (such as perceived ability, coping strategies, self-efficacy) and external "resources" (typically different types of social support) are particularly critical in building up resilience of the adolescents.

Among all five types of childhood maltreatment, resilience played a significant mediation role in their associations with SI only for EA, EN, and PN. Especially for EN and PN, as their direct associations with SI were completely mediated by resilience. After careful literature review, we have not identified any studies that attempted to discuss and compare the mediation of resilience in associations between different types of childhood maltreatment and SI among adolescents. However, a previous study found that, among all types of child abuse, ego-resiliency only significantly mediated the relationships between EA, PN, EN and three psychological symptoms (depression, self-harm behaviors, anxiety) (*Hong, 2017*). Emotional neglect in adolescents may lead to shame, behavioral avoidance, and a dysfunctional attachment style, which then will increase the risk of depression and anxiety, two recognized risk factors of SI (*Lee et al., 2018*).

Interpersonal assistance, suggested by our analytical results, was the strongest mediator in the association between childhood neglect (either EN or PN) and SI among the 5 dimensions of resilience measured in RSCA. For children and adolescents, the most

prominent interpersonal relationship involves friends at school (*Eccles & Roeser, 2011*). Therefore, interpersonal assistance that we measured in this study was largely peer support. In fact, friends and peers are critical in the development of social relationships and interpersonal connectedness during the stage when children and adolescents begin to form bonds outside of their families (*Gorrese & Ruggieri, 2012*). Thus, close friends and peers logically become the primary source of intimacy and social support at this stage of life (*Wilkinson, 2004*). Peer support has been identified as a protective factor against psychological problems, especially depression, among children and adolescents (*Stice et al., 2011*; *Mizuta et al., 2017*). *Yearwood et al. (2019)* have found that higher level of peer support could mitigate internalizing and externalizing psychopathological symptoms associated with abuse and neglect experience in adolescents. Besides, by surveying 8,778 Chinese adolescents, *Cui et al. (2010)* observed that the lack of peer support was significantly related to increased risk of SI. However, it is highly possible that the relationship between interpersonal support and SI can be inverse, as our analysis was based on cross-sectional data, and this possible inverse association can also be supported by previous publications (*Wiklander, Samuelsson & Asberg, 2003*). Nevertheless, under any situation, the importance of interpersonal assistance, especially peer support, should be recognized when designing and implementing SI prevention strategies for adolescents who had experienced childhood neglect.

Another important finding is that, for all dimensions of resilience, emotion regulation presented the strongest mediation in the association between EA and SI. Besides, emotional regulation also played a salient role in mediating the relationship between child neglect and SI. All these findings suggest that improving emotion regulation ability can be another promising way to help adolescents suffering from EA, EN and PN antagonize the risk of SI. Fortunately, some effective intervention programs targeting at optimizing emotion regulation and emotional competence among adolescents have already been implemented, such as Mastering emotions technique (MEMT) (*Patel, Nivethitha & Mooventhan, 2018*), emotional schema therapy (*Bradley et al., 2011*), Integrative Body-Mind Training (IBMT) (*Tang, Tang & Posner, 2016*). However, as they seldomly been used in Chinese adolescents, their effectiveness in preventing child abuse related SI should be further discussed.

The following limitations of our study should be noticed. At first, all our analytical results were based on cross-sectional data, therefore, our major findings, especially the positive mediation of resilience in maltreatment-SI association are to be further corroborated by future longitudinal studies with large sample sizes. Another limitation is that, as we only adjusted for limited confounders when fitting the path model, residual confounding will exist inevitably. Besides, childhood maltreatment was ascertained by self-reporting, so the possibility of recall bias and information covering caused by stigma cannot be eliminated. Finally, we chose study participants from a single city of Yunnan province, as they may not be representative to the entire Chinese children and adolescent population, the extrapolation of study results should be made with caution.

Despite all limitations stated above, our study is among the first attempts in investigating the mediation of resilience in the association between childhood maltreatment and SI in Chinese adolescents. The large sample size provides further consolidation to the validity

of the results. Our major findings suggest that building up psychological resilience, especially strengthening emotion regulation skills and consolidating interpersonal support, might be useful in reducing the risk of future suicide among Chinese adolescents who were childhood maltreatment victims. The long-term effect of resilience in mediating the association between childhood maltreatment and SI is to be corroborated by future prospective studies.

## CONCLUSIONS

In conclusion, this population-based cross-sectional study revealed a strong association between childhood maltreatment and SI, moreover, resilience played as a positive mediator in this association. Among the five dimensions of resilience, emotion regulation and interpersonal assistance presented the strongest mediation. Our findings suggested that resilience-oriented strategies could be effective in reducing the risk of childhood maltreatment related suicidal behaviors among Chinese children and adolescents, especially intervention measures focusing on strengthening emotion regulation skills and consolidating social support.

### Funding
This work was supported by the National Natural Science Foundation of China (No. 82060601), the Yunnan Applied Basic Research Projects-Kunming Medical University Union Foundation (No.2018FE001(-132)), the Yunnan Health Training Projects of Highly Level Talents (No. D-2017048), the Research Project of Medical and Sanitary Institution of Yunnan Province (No. 2018NS0110), the Top Young Talents of Yunnan Ten Thousand Talents Plan (No. qYNWR-QNBJ-2018-286) and the Innovative Research Team of Yunnan Province (No. 202005AE160002). The funders had no role in study design, data collection and analysis, decision to publish, or preparation of the manuscript.

### Grant Disclosures
The following grant information was disclosed by the authors:
National Natural Science Foundation of China: No. 82060601.
Yunnan Applied Basic Research Projects-Kunming Medical University Union Foundation: No.2018FE001(-132).
Yunnan Health Training Projects of Highly Level Talents: No. D-2017048.
Research Project of Medical and Sanitary Institution of Yunnan Province: No. 2018NS0110.
Top Young Talents of Yunnan Ten Thousand Talents Plan: No. qYNWR-QNBJ-2018-286.
Innovative Research Team of Yunnan Province: No. 202005AE160002.

### Competing Interests
The authors declare there are no competing interests.

## Author Contributions

- Xue Chen and Linling Jiang performed the experiments, analyzed the data, prepared figures and/or tables, authored or reviewed drafts of the paper, and approved the final draft.
- Yi Liu, Hailiang Ran, Runxu Yang and Xiufeng Xu performed the experiments, prepared figures and/or tables, and approved the final draft.
- Jin Lu and Yuanyuan Xiao conceived and designed the experiments, prepared figures and/or tables, authored or reviewed drafts of the paper, and approved the final draft.

## Ethics

The following information was supplied relating to ethical approvals (i.e., approving body and any reference numbers):

The Third People's Hospital of Lincang Ethics Committee approved this research (2019-01).

## Data Availability

The raw data are available in the Supplemental File.

## Supplemental Information

Supplemental information for this article can be found online at http://dx.doi.org/10.7717/peerj.11758#supplemental-information.

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
