# Peer review of "Childhood maltreatment and suicidal ideation in Chinese children and adolescents: the mediation of resilience"

_PeerJ, doi:10.7717/peerj.11758_

## Round 0.1 · original submission · Major Revisions

In addition to the comments of the two authors, the authors need to consider the below concerns.

1. Line 83-84, the authors suggested an indirect role, but how does this indicate a mediating role? The authors need to explain why resilience was hypothesized as a mediator on maltreatment-SI association.

2. Line 158-159, the authors mentioned moderation and production term, but in the title the authors indicated mediation. The two terms are different. Please specify the current study focused on mediating or moderating effects.

3. In the title, the authors focused on mediating effect, but in the main text, they examined factors associated with SI by using a lot of paragraphs. Please explain the objectives of this study and revise the necessary parts accordingly.

4. In clinical research methodology, cross-sectional data are not feasible for mediation analysis.

Reviewer 1 ·

Basic reporting

None

Experimental design

None

Validity of the findings

None

Additional comments

This is a population-based cross-sectional survey, which aimed to estimate the possible associations between childhood maltreatment and suicidal ideation in a large group of Chinese adolescents. Though this is a well-organized manuscript, I have some comments for the authors for their consideration.
1. The manuscript mentioned that the purpose of this study is to explore the possible mediation. However, I found you examined the moderation in the result section. Why not refer to the moderating effect in the introduction?
2. The manuscript mentioned that “47 subjects refused to participate”. Authors should provide the reasons.
3. Authors should provide the references about “The Resilience Scale for Chinese Adolescents (RSCA)”.
4. The manuscript mentioned “Data were analyzed as previously described in Xiao et al., 2019.” Some questions for this sentence were as follow.
1. Please use “analyzed” to replace “analysised”.
2. What the kind of data was analyzed in Xiao et al., 2019?
3. I checked the articles from Xiao et al., 2019. However, the population in Xiao et al., 2019 focus on the “left-behind” children and adolescents. So, does the population in this study mainly included the “left-behind” children and adolescents?
5. Table 1 should provide P values for univariate analysis between SI and non-SI.
6. Line 184. Authors used the median of RSCA (94) to dichotomize study participants. Please add related references.

·

Basic reporting

no comments

Experimental design

no comments

Validity of the findings

Query # 1: In table 2, there were some significant variables such as ‘Grade’, ‘Study style’ and ‘Marital status of the parents’ for SI . The authors should list these positive results in the section of Results.

Query # 2: In Figure 3, the authors should make clear the path coefficient ‘-0.441’ between ‘Resilience’ and ‘SI’. As figure 3 were results of a group of path analysis, I’m not sure the source of the coefficient ‘-0.441’.

Additional comments

The study was well organized and presented in a proper way. The study was a meaning effort to explore relationship among resilience, childhood maltreatment and suicidal ideation in a Chinese adolescent sample. Further longitudinal and some larger sample of studies are merited to explore validity of results of the study.

---

## Round 0.2 · Major Revisions

First of all, the English language of the paper remains poor, so the authors must have their paper edited after extensive revisions.

Second, the title is not accurate even problematic. I do not know why the authors preferred to use the term “indirect role” of resilience. In fact, the mediating role of resilience indicates a direct effect of reliance on SI.

Third, the paper was still written and organized in a confused way. Please revise it substantially.

Abstract. In the part of method, please briefly describe the assessment of childhood maltreatment, resilience, and SI, because they are important. The statistical methods were described in a confused way because in the background part, the authors declared to test mediating effect, but here they also tested moderation.

Line 47, WTO is very unusual which should be WHO. Further, the data cited here were too old. Please cited up-to-date statistics.

Line 74-77, these suggested a mediating role of resilience only. I can not see any supporting possibility for the moderating role of resilience.
The “indirect role” of resilience is not real indirect role. The authors must correct this all throughout the paper.

The authors must have a strong theoretical hypothesis for the mediating role of resilience and tested in the statistical analysis accordingly. The current analysis on the moderating role of resilience is completely data-driven, which is not convincing.

Reviewer 1 ·

Basic reporting

None

Experimental design

None

Validity of the findings

None

Additional comments

I am satisfied with your revision. The minor problems still need to be addressed.
(1) For easy identification, please add a comma to thousands of numbers in the whole manuscript. For example, Please use "3,146" in the line 27.
(2) Please keep a consistent format between Line 195-197 and Line 204.
(3)Please add the footnote for OR and CI in the table 2 and Figure 2.

·

Basic reporting

in line 202, 'aOR' should be a misspelling. Please check it.

Experimental design

no comments.

Validity of the findings

no comments.

Additional comments

The study was presented in a more proper way after its first revision.

---

## Round 0.3 · accepted · Accept

The authors have addressed the remaining concerns.

Reviewer 1 ·

Basic reporting

None

Experimental design

None

Validity of the findings

None

Additional comments

None